# Leukocyte–Cancer Cell Fusion—Genesis of a Deadly Journey

**DOI:** 10.3390/cells8020170

**Published:** 2019-02-18

**Authors:** Greggory S. Laberge, Eric Duvall, Kay Haedicke, John Pawelek

**Affiliations:** 1Human Medical Genetics and Genomics Program, University of Colorado School of Medicine, Aurora, CO 80045, USA; Greggory.Laberge@UCdenver.edu; 2Denver Police Crime Lab-Forensics and Evidence Division, Denver, CO 80204; Eric.Duvall@denvergov.org; 3Department of Internal Medicine Section of Medical Oncology and the Yale Cancer Center, Yale School of Medicine, New Haven, CT 06520-8059, USA; kay.haedicke@yale.edu; 4Department of Dermatology and The Yale Cancer Center, Yale School of Medicine, New Haven, CT 06520-8059, USA

**Keywords:** leukocyte–cancer cell fusion, metastasis, new therapeutic targets

## Abstract

According to estimates from the International Agency for Research on Cancer, by the year 2030 there will be 22 million new cancer cases and 13 million deaths per year. The main cause of cancer mortality is not the primary tumor itself but metastasis to distant organs and tissues, yet the mechanisms of this process remain poorly understood. Leukocyte–cancer cell fusion and hybrid formation as an initiator of metastasis was proposed more than a century ago by the German pathologist Prof. Otto Aichel. This proposal has since been confirmed in more than 50 animal models and more recently in one patient with renal cell carcinoma and two patients with malignant melanoma. Leukocyte–tumor cell fusion provides a unifying explanation for metastasis. While primary tumors arise in a wide variety of tissues representing not a single disease but many different diseases, metastatic cancer may be only one disease arising from a common, nonmutational event: Fusion of primary tumor cells with leukocytes. From the findings to date, it would appear that such hybrid formation is a major pathway for metastasis. Studies on the mechanisms involved could uncover new targets for therapeutic intervention.

## 1. Introduction

Several years ago, our group became attracted to a proposal published in 1911 by a German pathologist, Prof. Otto Aichel, that metastasis might result from the fusion between motile leukocytes and cancer cells, with the qualitative differences between chromosomes causing the hybrid to be ‘‘thrown out of the path of the mother cells to form what has come to be known as a malignant cell and resulting in an entirely new cell, having the characteristics of both mother cells” [1]. In this prescient statement, Aichel not only provided an explanation for metastasis but he also predicted the science of cancer epigenetics. That is, a new hybrid cell with characteristics of both “mother cells” in today’s terminology would refer to gene expression patterns from both fusion partners in the same cell. For example, at least some hybrids would express the leukocyte traits of motility, chemotaxis, and homing while at the same time have the uncontrolled cell division of the cancer cell as well as immuno-markers from both partners. To investigate this concept, our group has been studying cancer patients who had previously received an allogeneic bone marrow transplant (BMT), usually for leukemia or lymphoma, and then later developed a solid tumor. By analyzing tumor cells for both donor and patient DNA, we reasoned that such cells were likely to be leukocyte-tumor cell hybrids.

(i). Leukocyte–cancer cell fusion and hybrid formation in a renal cell carcinoma detected through the use of fluorescence in situ hybridization (FISH).

In our first case, we studied a primary renal cell carcinoma from a female patient who, two years prior to detection of the tumor, had received a BMT from her son. Due to the male donor–female recipient nature of the BMT, FISH could be used to search for putative BMT–tumor hybrids [2]. Karyotyping revealed that the tumor cells contained a clonal trisomy 17. Using dual-label FISH, the donor Y and three or more copies of chromosome 17 were visualized together in individual nuclei of carcinoma cells, providing direct genetic and morphological evidence for BMT–tumor hybrids (Figure 1). For example, Panel A shows a cell with three copies of chromosome 17 (green) but no Y chromosome, indicating that this cell was likely not a hybrid, while Panel B shows a trisomy 17 (green) plus the Y chromosome (red), indicating that the cell was a hybrid between a patient and a male donor cell. Such cells were in abundance in an area covering about 10% of the tumor, suggesting a clonal origin of the hybrids. One problem in the interpretation of these results is the phenomenon of fetal michrochimerism. Microchimerism usually concerns fetal cells in the mother’s circulatory system and elsewhere that were acquired during pregnancy [3]. For example, during pregnancy, fetal microchimerism can be sought from the mother’s blood for the purpose of prenatal diagnosis [4]. Thus in theory, the cell in Figure 1A could have been a cell from the male fetus containing a trisomy 17 wherein the Y chromosome was lost, while Figure 1B could have been another such cell wherein the Y chromosome was not lost. While this scenario is possible, we feel it is quite unlikely that the male cell would have lost its Y and that the explanation of fusion and hybridization is by far the most likely.

(ii). The first detection of leukocyte–cancer cell fusion and hybrid formation in a patient with melanoma using forensic short tandem repeat (STR) length polymorphisms to distinguish donor and patient genomes.

The first evidence for leukocyte–cancer cell hybrids in a human using DNA genotyping methods came from our study of a patient who had received an allogeneic BMT for lymphoma and later developed a melanoma brain metastasis with a donor–patient hybrid genome [5]. Tumor cells were isolated by laser microdissecton and sections were analyzed throughout the tumor, using forensic short tandem repeat (STR) length polymorphisms to distinguish donor and patient genomes. Tumor and pretransplant blood lymphocyte DNAs were analyzed for donor and patient alleles at 14 autosomal STR loci and the sex chromosomes. Eight of these loci were informative and indicated the presence of donor–patient hybrids. Figure 2 shows these loci with peaks from the electropherograms designated by asterisks with the following colors: black (donor and patient), red (donor only), and blue (patient only). Both donor and patient alleles were present in tumor cells throughout the tumor (sample numbers) and the tumor appeared to consist largely if not solely of bone-marrow-derived cell (BMDC)–tumor cell hybrids. Moreover, similar allelic ratios for each locus in sections throughout the tumor indicated a clonal origin of the metastasis and suggested that the tumor was generated from a prior fusion event between a single donor BMDC and patient tumor cell. We therefore conclude that the tumor-initiating cell was a BMDC–tumor cell hybrid.

(iii). The second detection of leukocyte–cancer cell fusion and hybrid formation in a patient with melanoma using forensic short tandem repeat (STR) length polymorphisms to distinguish donor and patient genomes.

Using fragment-length STR analyses, the second evidence for leukocyte–cancer cell hybrids came from a man who, eight years following an allogeneic BMT from his brother for treatment of chronic myelogenous leukemia (CML), developed a nodular malignant melanoma on the upper back with spread to an axillary sentinal lymph node [6]. Combining laser microdissection with detection of STR length polymorphisms, we were able to distinguish donor and patient genomes. Tumor and pretransplant blood lymphocyte DNAs were analyzed for donor and patient alleles at 15 autosomal STR loci and the sex chromosomes. DNA analyses of the primary melanoma and the nodal metastasis revealed that they exhibited alleles at each STR locus that were consistent with both the patient and donor. The doses varied between these samples, indicative of the relative amounts of genomic DNA derived from the patient and donor. Table 1 shows genotyping results using short tandem repeats (STRs) at each of the loci of DNA from donor (D), patient (P), primary tumor, and lymph node metastasis. As with the prior cases, the evidence supports fusion and hybridization between donor and patient cells as the initiator of metastasis in this patient.

Several laboratories have entered this field and extended the above findings significantly [7]. For example, Berndt et al. showed that fusion between leukocytes and normal cells can induce aneuploidy and drug resistance. After fusion (e.g., between a leukocyte or fibroblast and a cancer cell), the cell contains two nuclei, one from each fusion partner. When nuclei fuse into one, there is a random loss of chromosomes such that individual hybrid cells can have different genotypes. The aneuploidy may increase with subsequent cell divisions. Further, mutations in the cells may occur during these processes wherein some of the tumor cells could acquire drug and radiation resistance such that when such therapies are applied to the patient, the resistant cells survive and unfortunately form new and far more deadly tumors. Today, cancer cell resistance to various therapies remains a central problem in patient survival [8]. Using the Cre-loxP-based method, Searles et al. found that in cell culture, cancer cells can rapidly deliver DNA to macrophages and fibroblasts producing hybrids. Such cells were aneuploidy, exhibited increased clonal diversity, and acquired chemoresistance compared to nonhybrid cancer cells. Using reporter mice, they also observed the formation of hybrids between B16-GFP-Cre melanoma cells and normal cells in vivo. However, in the latter experiments, the frequency of hybrid formation was low and the cells did not appear to remain viable [9]. Mohr et al. investigated the factors and conditions through which plasma membranes between two different cells fuse, creating a single cell. They used a fluorescence double reporter vector (pFDR) containing a LoxP-flanked HcRed/DsRed expression cassette followed by an EGFP expression cassette to investigate four human breast cancer cell lines that had been stably transfected with a pFDR vector. They were cocultured with Cre-expressing human breast epithelial cell line. They then tested a panel of various cytokines, chemokines, growth factors, exomes, and other agents. Experiments were performed under both normoxic and hypoxic conditions. They found that the proinflammatory cytokine TNF-α under hypoxia is a potent inducer of cell fusion in human MDA-MB-435 and MDA-MB-231 breast cancer cells [10]. Sottile et al. investigated cell-to-cell interaction between mouse mesenchymal stem cells (MSCs) and embryonic stem cells (ESCs) and found that MSCs can either fuse, forming heterokaryons, or be invaded by ESCs through entosis. While the chromosomes from entosis-derived cells do not fuse into a single nucleus and in that sense are not true hybrids, the nuclei of fusion-derived cells can indeed hybridize into a single nucleus. Hetero-to-synkaryon transition occurs through cell division and not by nuclear membrane fusion. They also found that the ROCK-actin/myosin pathway is required for both fusion and entosis in ESCs but only for entosis in MSCs. Thus they show that MSCs can undergo fusion or entosis by generating distinct functional cells. The two processes are quite different. The authors conclude that and the outcomes should be considered when using MSC-based therapies [11]. Gast et al. recently developed methods for detecting hybrids in peripheral blood of human cancer patients that correlate with disease stage and predict overall survival. They pointed out that such hybrids might be used as biomarkers to assess disease progression [12].

## 2. Macrophage Traits in Metastatic Cancer Cells

In addition to direct genomic evidence for a relationship between leukocyte–cancer fusion and cancer progression are the large number of macrophage-like traits expressed by metastatic cancer cells. For example, Kemény et al. showed that melanoma cells spontaneously fusing with macrophages and fibroblasts in vitro can adopt the phenotypes of these cells [13].

Broncy and Paterlini-Bréchot reviewed evidence that circulating cancer cells expressed both epithelial and macrophage-specific markers [14]. These included CD14+/CD11c+ cells of myeloid lineage. B7-H4 is a cell surface antigen encoded by the VTCN1 gene, meaning V-set domain containing T cell activation inhibitor 1 which interacts with ligands bound to receptors on the surface of T cells and has been correlated with tumor progression. CD163 protein is a member of the scavenger receptor cysteine-rich superfamily and is exclusively expressed at the cell surface by monocytes and macrophages. CD146 refers to the Melanoma Cell Adhesion Molecule (MCAM) that is expressed in the cytoplasm of adipose and stromal progenitor cells. The CD68 protein is a transmembrane glycoprotein that is highly expressed by human monocytes and tissue macrophages. CD45 refers to the protein tyrosine phosphatase receptor type C (PTPRC) that is a transmembrane receptor expressed by mature leukocytes. The CD14 protein is a cell surface antigen expressed on monocytes and macrophages, but also present on other subtypes of myeloid cells such as dendritic cells. CD11b refers to the integrin subunit alpha M (ITGAM) and CD11c to the integrin subunit alpha X (ITGAX) that are both parts of leukocyte-specific integrins. CD133 refers to prominin 1, a transmembrane glycoprotein which localizes to membrane protrusions and is often expressed on adult stem cells, where it is thought to function in maintaining stem cell properties by suppressing differentiation. CD204 refers to the macrophage scavenger receptor 1 (MSR1) which is a macrophage-specific trimeric integral membrane glycoprotein. CD206 refers to the mannose receptor C-type 1 (MRC1) which is a type I membrane receptor that mediates the endocytosis of glycoproteins by macrophages. Cytokeratins (CK) are intermediate filaments expressed in epithelial tissues and are often used as a specific marker of epithelial cells. The epithelial cell adhesion molecule (EpCAM) is a membrane protein expressed on most normal epithelial cells that functions as a homotypic calcium-independent cell adhesion molecule. Vimentin is a type III intermediate filament protein which is responsible for maintaining cell shape and integrity of the cytoplasm in mesenchymal cells but has also recently been associated with tumor cells when expressed at the cell surface (i.e., cell surface vimentin, (CSV)) presenting with enlarged nuclei, CD45+ and exhibiting cytoplasmic staining by cytokeratins 8, 18, and 19 and epithelial cell adhesion molecule (EpCAM) [15]. Lustberg et al. reported about a population of circulating “atypical cells” expressing cytokeratins 8, 18, and 19, CD45 and CD68 markers without concomitant expression of EpCAM in the blood of metastatic breast cancer patients [16]. Recent reports about circulating atypical macrophages have now shed more light on these cells, on the possible mechanism of their formation, and on their relevance in tumor invasion. Earlier studies had pointed out the heterogeneous nature of circulating “atypical cells”, in particular regarding circulating tumor cells (CTCs), endothelial and epithelial cells, fibroblasts, macrophages, and megakaryocytes [17,18]. However, only cyto-morphological studies were possible at that time, as no histochemical characterization was available.

Chen et al. reported circulating macrophages expressing antigens expressed by tumor cells CD68 and B7–H4, in the blood of 56 lung cancer patients. They showed that CD68+ and B7–H4+ circulating macrophages correlated with tumor size and lymph node metastasis [19].

Adams et al. reported circulating atypical cells with concomitant expression of macrophage-specific and epithelial-cell-specific markers. They speculated that cancer-associated macrophage-like cells [CAMLs] might be different stages of myeloid differentiation and/or derive from nonspecific engulfment of epithelial cellular debris. They also described that some CAMLs bind to and migrate in blood attached to CTC [19].

Shabo et al. showed that macrophage traits in cancer cells are induced by macrophage-cancer cell fusion and cannot be explained simply by cellular interactions. They showed that tumor cell expression of the macrophage marker CD163 is related to poor prognosis in patients with breast cancer, colorectal cancer, and urinary bladder cancer [20,21,22,23,24,25].

Leukocyte–cancer cell fusion as a source of myeloid traits in cancer has also been discussed by Pawelek and colleagues [26,27,28,29,30,31,32,33,34]. Following macrophage–cancer cell fusion, the resultant hybrid cells acquired new abilities to promote angiogenesis, matrix alterations, motility, chemotaxis, and immune signaling pathways. Macrophage–tumor cell fusion could explain the aneuploidy, plasticity, and heterogeneity of malignant melanoma and it could also account for epidermal–mesenchymal transition in tumor progression since macrophages are of mesodermal origin [26]. There is considerable evidence that fusion between macrophages or other phagocytes and cancer cells causes epigenetic reprograming. Following fusion in vitro between weakly metastatic Cloudman S91 mouse melanoma cells and mouse or human macrophages, more than half of the resulting hybrids were more metastatic that the parental cell line. The metastatic hybrids showed increased expression of a number of macrophage-like molecules including SPARC, SNAIL, MET, MITF; integrin subunits α3, α5, α6, αv, β1, β3 [28], GnT-V (β1,6-acetylglucosaminyltransferase-V) and its enzymatic products β1,6-branched oligosaccharides conjugated to *N*-glycoproteins [27,28,29], cell-surface LAMP1 [30], high levels of autophagy [32], acquired hormone inducible chemotaxis [33], and expression of c-Met pro-oncogene [34]. These traits are all associated with tumor progression and poor outcome in a number of cancers [27].

Thus there is now considerable evidence from several sources that fusion and hybridization of phagocytes such as macrophages with cancer cells creates metastatic cells. Our group has demonstrated this in two patients with melanoma and one with renal cell carcinoma. In addition, several labs have made immunological observations that metastatic cancer cells exhibit many macrophage traits. Thus it seems safe to say that this is at least one mechanism for metastasis. This confirms the century-old proposal of Prof. Otto Aichel that in retrospect was prescient indeed, especially considering that he had only a microscope with which to work.

For the first time, we can glimpse the engine that drives metastasis. A scheme for this is shown in Figure 3. This information opens many potential targets for the development of new therapies, for example: (1) inhibition of the fusion process itself regarding events such as membrane attachment and heterokaryon formation; (2) inhibition of the hybridization processes involving integration of parental fusion partner genes into hybrid genomes; and (3) prevention of post-hybridization events involving activation of genes that control cell migration, chemotaxis, intravasation, extravasation, and migration to lymph nodes and distant metastases.

## Figures and Tables

**Figure 1 cells-08-00170-f001:**
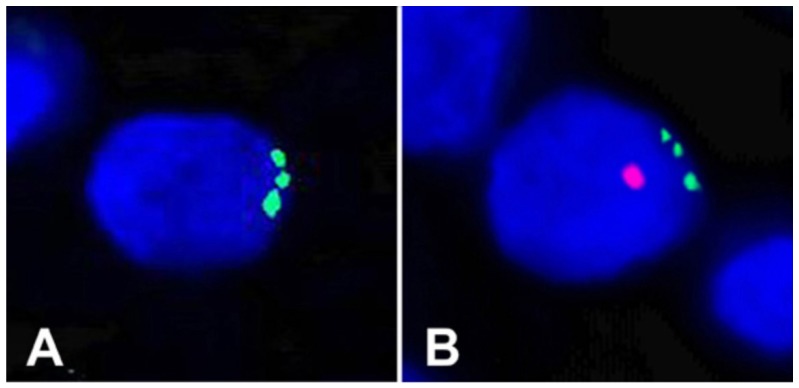
FISH analyses of formalin-fixed sections of a primary renal cell carcinoma described herein. The slides were counter-stained with DAPI [2]. A primary renal cell carcinoma from a female patient who, two years prior to detection of the tumor, had received a BMT from her son. Due to the male donor–female recipient nature of the BMT, FISH could be used to search for putative BMT–tumor hybrids [2]. Karyotyping revealed that the tumor cells contained a clonal trisomy 17. Using dual-label FISH, the donor Y and three or more copies of chromosome 17 were visualized together in individual nuclei of carcinoma cells. Panel (**A**) shows a cell with three copies of chromosome 17 (green) but no Y chromosome, indicating that this cell was likely not a hybrid, while Panel (**B**) shows a trisomy 17 (green) plus the Y chromosome (red), indicating that the cell was a hybrid between a patient and a male donor cell.

**Figure 2 cells-08-00170-f002:**
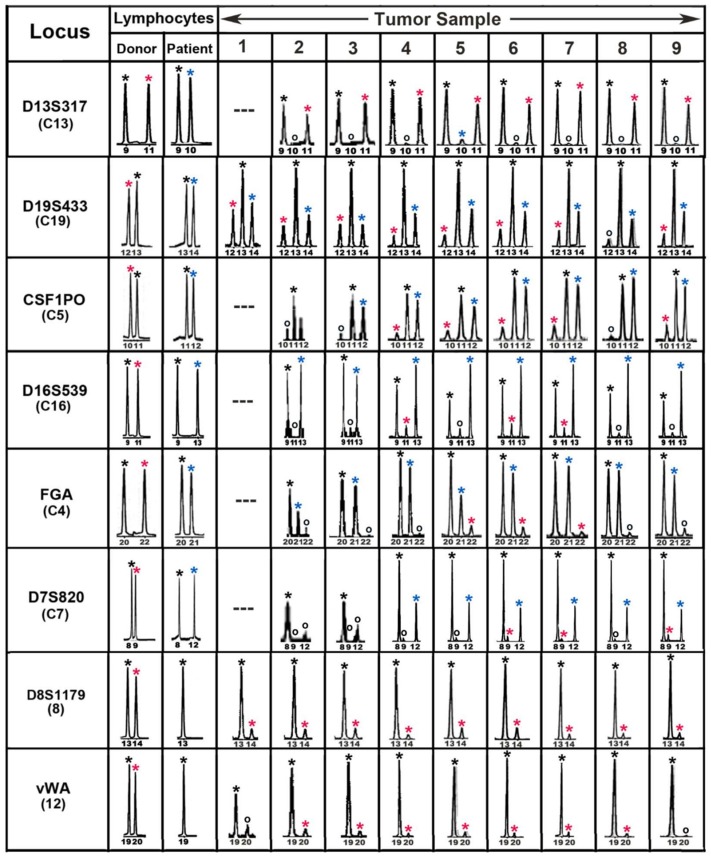
Forensic STR analyses of the MH3 melanoma along with donor and patient pre-BMT lymphocytes. Shown are ‘‘informative’’ loci exhibiting donor- and patient-specific alleles in pre-BMT lymphocytes. Tumor loci are listed in order of relative abundance of the donor-specific alleles (red asterisk) compared to patient-specific (blue asterisk) and shared alleles (black asterisk). Allele peaks, 50 relative fluorescence units were censored as ‘‘no call’’ (open circles). Loci with no detectable alleles after PCR amplification (—) [5].

**Figure 3 cells-08-00170-f003:**
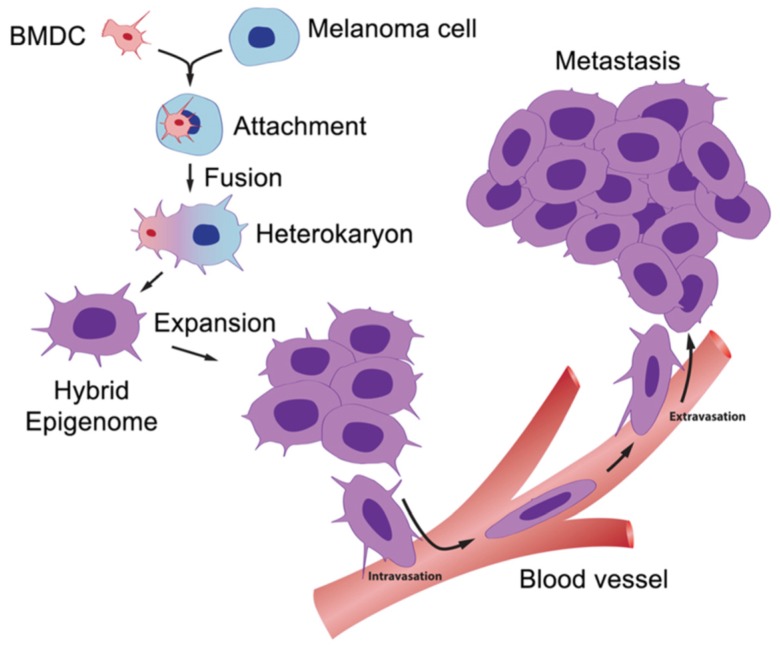
The BMDC–cancer cell fusion hypothesis. A motile BMDC (red), such as a macrophage or stem cell, is drawn to a cancer cell (blue). The outer cell membranes of the two cells become attached. Fusion occurs with the formation of a binucleated heterokaryon having a nucleus from each of the fusion partners. The heterokaryon goes through genomic hybridization creating a melanoma–BMDC hybrid with two gene expression patterns conferring deregulated cell division and metastatic competence to the hybrid [5].

**Table 1 cells-08-00170-t001:** STR genotyping of DNA from donor (D), patient (P), primary tumor, and lymph node metastasis. STR units: number of tandem repeats of the locus-specific tetranucleotide sequence [6].

STR Locus	Primary Tumor	Lymph Node	Patient Sample	Donor Sample
D8S1179	13, 15	13, 15	13, 15	13
D21S11	28, 29, 30, 30.2	28, 29, 30, 30.2	28, 29	30, 30.2
D7S820	11, 12	11, 12, 14	12, 14	11
CSF1PO	9, 11	10, 11, 12	11, 12	9, 10
D3S1358	15, 16, 18	16, 18	16, 18	15, 16
TH01	6, 7, 9	6, 7, 9	6	7, 9
D13S317	8, 9, 12	8, 12	12	8, 9
D16S539	13	11, 13	11, 13	13
D2S1338	16, 17, 18	17, 19	17, 19	16, 18
D19S433	13, 15, 16	13, 15, 16	15, 16	13, 16
vWA	17, 18, 19	17, 18	17, 18	18, 19
TPOX	8, 9, 11	8, 9	8, 9	8, 11
D18S51	12	12, 20	12, 20	15, 18
Amelogenin	X, Y	X, Y	X, Y	X, Y
D5S818	9, 11, 12	9, 11, 12	11	9, 12
FGA	21, 22, 24	21, 24	21, 24	22, 25

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
