# Peer review of "Leukocyte–Cancer Cell Fusion—Genesis of a Deadly Journey"

_cells, 2019, doi:10.3390/cells8020170_

Round 1
Reviewer 1 Report
The investigators have been leaders in the field championing the idea that fusions occur and that they may participate in metastases. However, this review is almost exclusively biased to their own work, with very little discussion of the work of others (just a few lines are devoted to this, generally with just a single phrase of discussion).
I think there are probably few other investigators (if any) working in the field who would agree "that it is now safe to say that this is the major if not sole mechanism for metastasis".
The clearly documented examples of fusions in humans seem to have all occurred in bone marrow transplant recipients, where immune function has been markedly disrupted.
Because fusions are present in focal lesions (which are presumably metastatic foci) does not imply that they established the lesions. Fusion appears to be an ongoing process, and they may have originated there. More importantly, if fusions are solely responsible for metastasis, one has to wonder why metastatic lesions are not composed solely of fusions. The literature clearly shows that this is not the case.
I definitely think that either much more discussion is needed, or these statements need to be modified for the reader to get a more balanced view of things.
Author Response
Reviewer #1
1. The investigators have been leaders in the field championing the idea that fusions occur and that they may participate in metastases. However, this review is almost exclusively biased to their own work, with very little discussion of the work of others (just a few lines are devoted to this, generally with just a single phrase of discussion).
Ans: The references to other people’s work have been expanded considerably
I think there are probably few other investigators (if any) working in the field who would agree "that it is now safe to say that this is the major if not sole mechanism for metastasis".
We completely agree and the phrase has been changed to “Thus it seems safe to say that this is at least one mechanism for metastasis.”
Reviewer 2 Report
This manuscript describes authors’ discoveries in the Introduction section, and the other related discoveries are described in only 3 paragraphs in the following section of “Macrophage traits in metastatic cancer cells”. These descriptions are too short to overview the hypothesis originally proposed by Prof. Otto Aichel. I think more detailed descriptions are needed as a review manuscript. A scheme showing the overview of the progression of tumor metastasis by cell fusion with leucocytes will help to understand the hypothesis. Also, following descriptions (these are just examples) might help to overview the hypothesis and the importance of authors’ discovery. More detailed summary for Prof. Otto Aichel’s hypothesis. History of the discovery for leucocyte-cancer cell fusion (in vitro, in vivo, animal model and human etc...) Experimental evidences showing that cell fusion with leucocyte initiates cancer metastasis and possible mechanisms for tumor metastasis by cell fusion.Author Response
The clearly documented examples of fusions in humans seem to have all occurred in bone marrow transplant recipients, where immune function has been markedly disrupted.
Ans: At the moment, studying patients with allogeneic BMTs is the only way to document hybrids because donor and recipient DNA’s can be distinguished from one another. Often these patients have compromised immune systems from radiation and chemotherapy and this appears to be the cause of the secondary malignancies. See the 3 refs below that are now in the manuscript. There are many more.
Refs:
-Yosuke Tanaka, Saiko Kurosawa, Kinuko Tajima, Yoshihiro Inamoto, Reiko Ito, Takashi Tanaka, Yoshitaka Inoue, Takayuki Ozawa, Akihisa Kawajiri, Akio Onishi, Keiji Okinaka, Shigeo Fuji, Sung-Won Kim, Ryuji Tanosaki, Takuya Yamashita and Takahiro Fukuda. Secondary Cancer after Allogeneic Hematopoietic Stem Cell Transplantation. Blood 2015 126:3162.
-Navneet S. Majhail, Ruta Brazauskas, J. Douglas Rizzo, Ronald M. Sobecks, Zhiwei Wang,, Mary M. Horowitz, Brian Bolwell, John R. Wingard, and Gerard Socie Secondary solid cancers after allogeneic hematopoietic cell transplantation using busulfan-cyclophosphamide conditioning Blood. 2011 Jan 6; 117(1): 316–322. PMCID: PMC3037753 PMID: 20926773
-Children (Basel). 2015 Jun; 2(2): 146–173.PMCID: PMC4928755 PMID: 27417356 Simon Bomken1,† and Roderick Skinner Secondary Malignant Neoplasms Following Haematopoietic Stem Cell Transplantation in Childhood
Reviewer 3 Report
Comments to Ms LaBerge et al.
LaBerge et al. give a short overview about Leukocyte-Cancer Cell Fusion and the Genesis of a Deadly Journey. In brief, this is an interesting manuscript and I only have a few comments.
Major comments
1) I have one comment to the data shown in Figure 1. In Panel B a cell with 3 copies of chromosome 17 and 1 Y chromosome is shown and the authors concluded that this cell is hybrid between a patient and a male donor cell. I think, that the authors do know about the phenomenon of fetal cell microchimerism. Because the female patient received a BMT from her son it cannot be ruled out completely that the cell shown in Panel B might be rather of fetal origin than of bone marrow origin. It would be great if the authors could comment on this.
2) On page 4, line 91-93: reference [6] is not correct since Berndt et al. have not used a Cre-LoxP-based double fluorescence reporter system.
Minor comments
1) page 1, line 9: please check spelling for the email address of Kay Haedicke (or Heidicke).
2) page 5, line 156: please remove the empty line
Author Response
Reviewer #3
Comment on microchimerism and Yilmaz paper DONE
LaBerge et al. give a short overview about Leukocyte-Cancer Cell Fusion and the Genesis of a Deadly Journey. In brief, this is an interesting manuscript and I only have a few comments.
Major comments
1) I have one comment to the data shown in Figure 1. In Panel B a cell with 3 copies of chromosome 17 and 1 Y chromosome is shown and the authors concluded that this cell is hybrid between a patient and a male donor cell. I think, that the authors do know about the phenomenon of fetal cell microchimerism. Because the female patient received a BMT from her son it cannot be ruled out completely that the cell shown in Panel B might be rather of fetal origin than of bone marrow origin. It would be great if the authors could comment on this.
Ans: A discussion of microchimerism is now included.
2) On page 4, line 91-93: reference [6] is not correct since Berndt et al. have not used a Cre-LoxP-based double fluorescence reporter system.
Ans: This has been corrected.
Minor comments
1) page 1, line 9: please check spelling for the email address of Kay Haedicke (or Heidicke).
Ans: This is now corrected
2) page 5, line 156: please remove the empty line
Ans: This is now corrected.
Round 2
Reviewer 1 Report
The work still comes across as biased toward the authors own work, but so be it.
Author Response
To the Reviewer,
In the first review I was asked to expand the paper to cite other authors in the field. It is now expanded considerably and I thank the reviewer for this because I feel it is much stronger. It remains true that several papers from our own group are cited, particularly in section 2, "Macrophage traits in metastatic cancer cells", but our work on this stems back some 20 years and to me it is important to include them.
Thank you for your help and I hope all is well now.
John Pawelek
Reviewer 2 Report
I am confusing what does the authors’ answer mean. Then I would like to make a few comments on the revised manuscript. The authors have added some detailed descriptions about related studies to the introduction. I think the introduction should be divided into several parts with a proper chapter title. There is no description about figure 3 in the main text.
Author Response
REVIEWER 2: I am confusing what does the authors’ answer mean. Then I would like to make a few comments on the revised manuscript. The authors have added some detailed descriptions about related studies to the introduction.
AU Reply #1
Reviewer 2 asked for more detail on the contributions of Prof Otto Aichel to cancer research. In the earlier draft we simply stated that he predicted the science of epigenetics. We have now enlarged this comment to read:
"Aichel not only provided an explanation for metastasis but he also predicted the science of cancer epigenetics. That is, a new hybrid cell with characteristics of both “mother cells” in today’s terminology would refer to gene expression patterns from both fusion partners in the same cell. For example, at least some hybrids would express the leucocyte traits of motility, chemotaxis and homing while at the same time have the uncontrolled cell division of the cancer cell as well as immuno-markers from both partners."
AU Reply #2
Reviewer 2 asked for the introduction to be divided into several parts with a proper chapter title.
The Introductions is now divided into sections
AU Reply #3
Reviewer 2 asked that a description of Figure 3 be entered into the main text.
Such a description has now been offered.